# Etiology and outcome of penetrating keratoplasty in bullous keratopathy post-cataract surgery vs post-glaucoma surgery

**Young-ho Jung[1,2], Hyuk Jin Choi[1,2], Mee Kum Kim[1,2], Joo Youn Oh[1,2]***

**1** Department of Ophthalmology, Seoul National University College of Medicine, Seoul, Republic of Korea, **2** Laboratory of Ocular Regenerative Medicine and Immunology, Biomedical Research Institute, Seoul National University Hospital, Seoul, Republic of Korea

* jooyounoh77@gmail.com, bonzoo1@snu.ac.kr

## Abstract

### Purpose

To investigate the causes of bullous keratopathy (BK) in the Korean population and analyze the results of penetrating keratoplasty (PK) in BK eyes associated with the top two causes: pseudophakic bullous keratopathy (PBK) and glaucoma surgery-associated BK (GBK).

### Methods

Medical records were reviewed of patients diagnosed with BK at a tertiary referral center between 2010 and 2020. The predisposing conditions, clinical characteristics and therapeutic outcomes after PK were analyzed and compared.

### Results

Of total 340 BK eyes, 70% (238 eyes) were associated with ocular surgery; most commonly, cataract surgery (48%, 162 eyes) and glaucoma surgery/laser (21%, 70 eyes). The BK onset was faster following glaucoma surgery/laser (91.7 ± 94.4 months) than following cataract surgery (160.7 ± 138.0 months, $p < 0.001$). The median survival time of allografts was shorter in GBK than in PBK (24.0 vs 51.0 months, $p = 0.020$). Best-corrected logMAR visual acuities were lower in GBK than in PBK after PK (1.4 ± 0.7 vs 0.9 ± 0.6, $p = 0.017$ at one year; 1.8 ± 0.7 vs 1.1 ± 0.8, $p = 0.043$ at three years).

### Conclusions

Intraocular surgery is the major predisposing condition of BK in Korea. GBK developed earlier and its therapeutic outcome was poorer, compared to PBK.

## Introduction

Bullous keratopathy (BK), caused by corneal endothelial dysfunction, is characterized by corneal stromal edema and often associated with bullae of the corneal epithelium. The edema

---

**Data Availability Statement:** All relevant data are within the manuscript.

**Funding:** This work was supported by the National Research Foundation of Korea (2021R1A2C3004532 to J.Y.O). The funders had

no role in study design, data collection and analysis, decision to publish, or preparation of the manuscript.

**Competing interests:** The authors have declared that no competing interests exist.

reduces the visual acuity (VA) and corneal epithelial bullae cause ocular pain. Thus, advanced cases of BK require treatment, and the definitive treatment is corneal transplantation to replace the dysfunctional corneal endothelium with healthy endothelium from a donor cornea. In fact, BK is one of the most common indications for corneal transplantation worldwide [1–4].

The etiologies that cause corneal endothelial dysfunction are diverse, including surgical/laser trauma, endothelial dystrophy, infection and immune-mediated damage [5]. Importantly, the causes of corneal endothelial dysfunction exhibit regional and chronological differences as indicated by a recent systematic review showing that reported indications for penetrating keratoplasty (PK) significantly vary [2]. For example, whereas Fuchs' endothelial corneal dystrophy (FECD) is the leading cause of visually significant corneal endothelial decompensation in western countries [1–5], it is rare in Asian countries, and an injury to the corneal endothelium during intraocular surgery is a more common cause of BK in an Asian eye having shallow anterior chamber (AC) and narrow angle [5, 6].

In this study, we investigated the predisposing conditions of BK in the Korean population between 2010 and 2020 and compared clinical characteristics of BK among different etiologies. Also, we comparatively analyzed the long-term results of PK in BK eyes associated with the top two etiologies: cataract surgery-associated BK (i.e. pseudophakic BK, PBK) and glaucoma surgery-associated BK (GBK).

## Patients and methods

The study was approved by the Institutional Review Board (IRB) of Seoul National University Hospital (IRB No. 2020-122-924), and conducted according to the principles expressed in the Declaration of Helsinki. Medical records were retrospectively reviewed of patients diagnosed with BK at our tertiary referral center (Seoul National University Hospital) between 2010 and 2020. Due to the retrospective nature of the study, the IRB waived the requirement for obtaining informed consent from the patients. Patients who had previously received PK before BK (i.e. BK due to corneal graft failure) and those with a follow-up period of < 3 months were excluded from analysis. As a result, a total of 340 BK eyes from 326 patients were included and analyzed in the study.

The following data were collected from medical charts: demographic information including age, gender and ethnicity/race, general medical history, ocular medical and surgical histories, the interval from a causative event to BK onset, ophthalmic findings including VA, intraocular pressure (IOP), lens status, endothelial cell density (ECD) and central corneal thickness (CCT), graft clarity at last follow-up (in cases with PK), and the follow-up period. The onset of BK was designated as the first date when the signs of BK were first observed in the medical record. The cause of BK was determined by two corneal specialists (Y.J. and J.Y.O.). Specifically, the diagnosis of PBK was made when BK developed after cataract extraction and intraocular lens (IOL) implantation in an eye with no evidence of corneal dystrophy or congenital anomaly and no history of glaucoma surgery/laser or ocular trauma. GBK was defined as BK that occurred following glaucoma surgery, such as glaucoma drainage device (GDD) implantation, trabeculectomy and trabeculotomy, irrespective of a history of cataract surgery. BK cases which received both cataract and glaucoma surgeries were designated as GBK.

For analysis of PK outcome, patients with the postoperative follow-up of ≥ 6 months were included. As a result, 5 out of total 114 eyes with PK were excluded from the outcome analysis due to insufficient follow-up duration (two PBK eyes, two GBK eyes and one uveitis-induced BK). Graft failure was defined as an irreversible loss of corneal graft clarity despite intensive medical treatment according to the cause of failure. Immunologic rejection of corneal grafts was defined as the sudden onset of corneal edema in the presence of ocular inflammation.

VA measured initially by Snellen charts was converted to logarithm of the minimum angle of resolution (logMAR) value for analysis. The improvement in VA was decided when best-corrected VA (BCVA) was improved by $\geq 0.3$ logMAR values [7, 8]. Patients with amblyopia, advanced glaucoma, or retinal disease (macular edema, retinal detachment and exudative age-related macular degeneration) significantly affecting VA were excluded from the visual outcome analysis. Increased IOP (IIOP) was defined as IOP > 21 mmHg as measured by Goldmann applanation tonometry, rebound tonometry or non-contact tonometry, and then was adjusted for corneal factors, including CCT and corneal curvature, using correction formulas such as the Ehlers and Doughty formulas [9, 10].

Statistical analysis was performed using SPSS Statistics 20.0 (IBM, Armonk, NY), and graphs were made using GraphPad Prism 8.4.2 (GraphPad Software, San Diego, CA). The chi-square test or Fisher's exact test was used for comparison of two categorical variables. Student's t-test was applied for comparison of mean values of continuous variables between two groups. The Kaplan-Meier curves were used to estimate the median time to BK onset or corneal graft failure, and the log-rank test was conducted to compare the differences between Kaplan-Meier curves. Multivariate Cox proportional hazards regression was used to assess multiple potential risk factors associated with graft survival. Data were presented as mean ± SD. Differences were considered significant at $p < 0.05$.

## Results

### Causes of BK

Totally, 340 BK eyes of 326 patients were included in the study. All were Koreans by ethnicity, including 126 women (38.7%) and 200 men (61.3%). The mean age at the time of BK diagnosis was 63.9 ± 14.9 years (1 − 87 years). Right eye was involved in 162 patients (49.7%), left eye in 150 patients (46.0%), and both eyes in 14 patients (4.3%).

The etiology of BK, clinical characteristics and PK outcomes according to each etiology are summarized in Fig 1 and Table 1. The major predisposing condition leading to BK was intraocular surgery or laser. Totally, 238 eyes (70%) of 340 BK eyes were associated with surgery or laser; cataract surgery (n = 162), glaucoma surgery (n = 48), vitrectomy (n = 6) and laser iridotomy (n = 22).

Cataract surgery was the most common cause of BK, responsible for 47.6% (n = 162) out of a total of 340 cases; the types of cataract surgeries included phacoemulsification (PE) (n = 109), extracapsular cataract extraction (ECCE) (n = 17), intracapsular cataract extraction (ICCE) (n = 22) and unknown (n = 14). Specifically, PBK accounted for 40.3% (n = 137), and aphakic BK (ABK) for 7.4% (n = 25). Among the 137 PBK eyes, 119 (86.9%) had a posterior chamber (PC) IOL, while 18 eyes (13.1%) had an AC IOL including implantable collamer lens (n = 2) and iris-claw IOL (n = 1). Additionally, 103 out of 137 PBK eyes occurred after PE, 13 after ECCE, and 13 after ICCE, while information on the type of cataract surgery was not available for the remaining 8 eyes.

The second leading cause of BK was glaucoma surgery/laser, representing 20.6% (n = 70) of total BK cases (n = 340); 14.1% (n = 48) occurred following glaucoma surgery (GDD implantation in 43 eyes, trabeculectomy in 3 and trabeculotomy in 2), and the other 6.5% (n = 22) followed laser iridotomy.

The third most common cause of BK was sterile or infectious inflammation, which accounted for 12.7% (n = 43) of total BK cases (n = 340); herpes simplex virus keratitis (n = 19), idiopathic anterior uveitis (n = 13), endophthalmitis (n = 4), cytomegalovirus endotheliitis (n = 3), toxic anterior segment syndrome (n = 2), varicella zoster virus endotheliitis (n = 1) and bacterial corneal ulcer (n = 1).

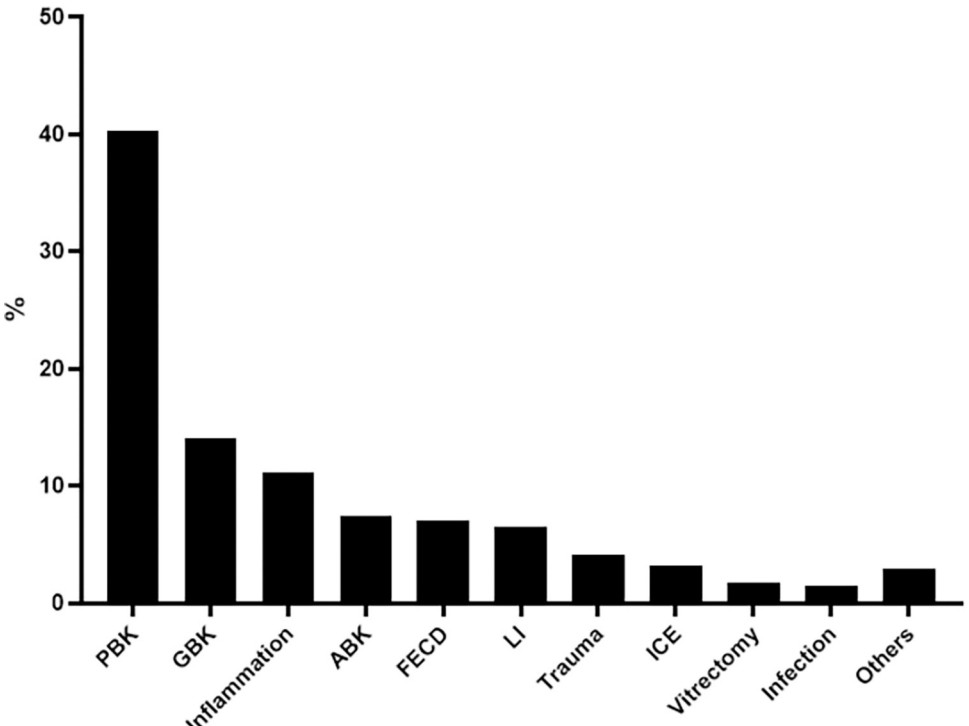

**Fig 1. Etiologies of BK in the Korean population.** The number of eyes in each etiology out of total 340 BK eyes is presented as %. The most common type of BK is PBK (n = 137, 40.3%), followed by GBK (n = 48, 14.1%).

**Table 1. Demographic profiles, clinical characteristics and PK outcomes of BK according to etiology.**

| | Eyes (n, %) | Age(years) | Sex (M:F) | Laterality (R:L:B) | OT (mo) | PK (n, %) | Mean ST (mo) | GF at last FU (n) | Post-PK FU (mo) |
|---|---|---|---|---|---|---|---|---|---|
| Total | 340 (100) | 63.9 ± 14.9 | 200:126 | 162:150:14 | 118.5 ± 129.1 | 114 (100) | 49.1 ± 4.1 | 62 | 53.9 ± 29.9 |
| PBK | 137 (40.3) | 65.6 ± 12.9 | 83:50 | 62:67:4 | 148.4 ± 125.1 | 51 (44.7) | 56.4 ± 6.6 | 26 | 57.9 ± 29.9 |
| ABK | 25 (7.4) | 63.5 ± 18.4 | 18:5 | 9:12:2 | 226.0 ± 184.7 | 8 (7.0) | 44.1 ± 11.6 | 6 | 68.0 ± 29.3 |
| PBK + ABK | 162 (47.6) | 65.2 ± 13.8 | 101:55 | 71:79:6 | 160.7 ± 138.0 | 59 (51.8) | 54.4 ± 8.1 | 32 | 59.3 ± 29.5 |
| GBK | 48 (14.1) | 62.9 ± 16.1 | 34:13 | 25:21:1 | 85.0 ± 94.6 | 14 (12.3) | 28.3 ± 5.9 | 8 | 50.0 ± 33.9 |
| LI | 22 (6.5) | 71.0 ± 6.4 | 5:16 | 13:7:1 | 107.7 ± 96.9 | 4 (3.5) | 50.3 ± 15.4 | 2 | 60.3 ± 21.1 |
| GBK + LI | 70 (20.6) | 65.5 ± 14.1 | 39:29 | 38:28:2 | 91.7 ± 94.4 | 18 (15.8) | 33.2 ± 9.3 | 10 | 52.6 ± 29.9 |
| Fuchs' dystrophy | 24 (7.1) | 67.8 ± 11.0 | 7:13 | 10:6:4 | N/A | 5 (4.4) | N/A | 0 | 27.4 ± 13.6 |
| ICE syndrome | 11 (3.2) | 61.1 ± 10.6 | 4:7 | 8:3:0 | N/A | 6 (5.3) | 26.3 ± 7.5 | 5 | 56.0 ± 29.8 |
| Inflammation* | 38 (11.2) | 62.5 ± 12.9 | 22:15 | 17:19:1 | 37.0 ± 58.8 | 9 (7.9) | 41.8 ± 6.7 | 4 | 37.6 ± 19.1 |
| Trauma | 14 (4.1) | 49.1 ± 17.2 | 12:2 | 5:9:0 | 111.0 ± 172.3 | 8 (7.0) | 48.7 ± 16.9 | 5 | 49.8 ± 39.4 |
| Vitrectomy | 6 (1.7) | 46.3 ± 14.3 | 6:0 | 3:3:0 | 7.8 ± 4.4 | 3 (2.6) | 21.3 ± 11.6 | 3 | 33.3 ± 18.9 |
| Infection† | 5 (1.5) | 75.2 ± 6.9 | 2:3 | 4:1:0 | 1.8 ± 0.8 | 1 (0.9) | 38 | 1 | 40 |
| Others | 10 (3.0) | 47.6 ± 25.7 | 8:2 | 7:3:0 | 47.6 ± 62.9 | 5 (4.4) | 57.7 ± 13.9 | 2 | 67.8 ± 27.9 |

BK: bullous keratopathy, M: male, F: female, R: right, L: left, B: bilateral, OT: onset time, mo: months, PK: penetrating keratoplasty, ST: graft survival time, GF: graft failure, FU: follow up, PBK: pseudophakic bullous keratopathy, ABK: aphakic bullous keratopathy, GBK: glaucoma surgery-associated bullous keratopathy, LI: laser iridotomy, ICE syndrome: iridocorneal endothelial syndrome, N/A: not available. *Inflammation includes herpes simplex virus keratitis (n = 19), idiopathic anterior uveitis (n = 13), cytomegalovirus endotheliitis (n = 3), toxic anterior segment syndrome (n = 2) and varicella zoster virus endotheliitis (n = 1). † Infection includes four endophthalmitis (n = 4) and one bacterial corneal ulcer (n = 1).

FECD was found to be the cause of BK in 7.1% (n = 24) of total BK cases (n = 340), followed by trauma (4.1%, n = 14), iridocorneal endothelial (ICE) syndrome (3.2%, n = 11) and vitrectomy (1.7%, n = 6, 5 of which had silicone oil injection). Among 14 BK eyes secondary to trauma, 9 were inflicted by blunt trauma, and 5 by penetrating injury.

Of 10 eyes (3.0%) classified as others, 5 were associated with high IOP, one with posterior polymorphous corneal dystrophy, one with retinopathy of prematurity, one with congenital glaucoma and one with chronic retinal detachment.

## Onset of BK

Further, we investigated the time to BK onset since the causative incident except in eyes where the disease had insidious course as in FECD and ICE syndrome. The results are presented in Table 1 and Fig 2.

The mean interval from the causative event to corneal edema was shortest in BK occurring after infection (1.8 ± 0.8 months) or vitrectomy (7.8 ± 4.4 months), followed by BK occurring after inflammation (herpes simplex virus keratitis, cytomegalovirus or varicella zoster virus endotheliitis, idiopathic anterior uveitis or toxic anterior segment syndrome) (37.0 ± 58.8 months). The onset time was longest in BK eyes associated with cataract surgery (160.7 ± 138.0 months); 148.4 ± 125.1 months for PBK and 226.0 ± 184.7 months for ABK. BK developed 85.0 ± 94.6 months after glaucoma surgery, 107.7 ± 96.9 months after laser iridotomy, and 111.0 ± 172.3 months after ocular trauma. Specifically, the mean time of BK onset was significantly shorter in GBK than in PBK (85.0 ± 94.6 months vs 148.4 ± 125.1 months, $p < 0.001$). Similarly, the median time to BK onset, as analyzed by Kaplan-Meier survival curves, was significantly shorter in GBK than in PBK (60 months vs 120 months, $p = 0.003$) (Fig 2).

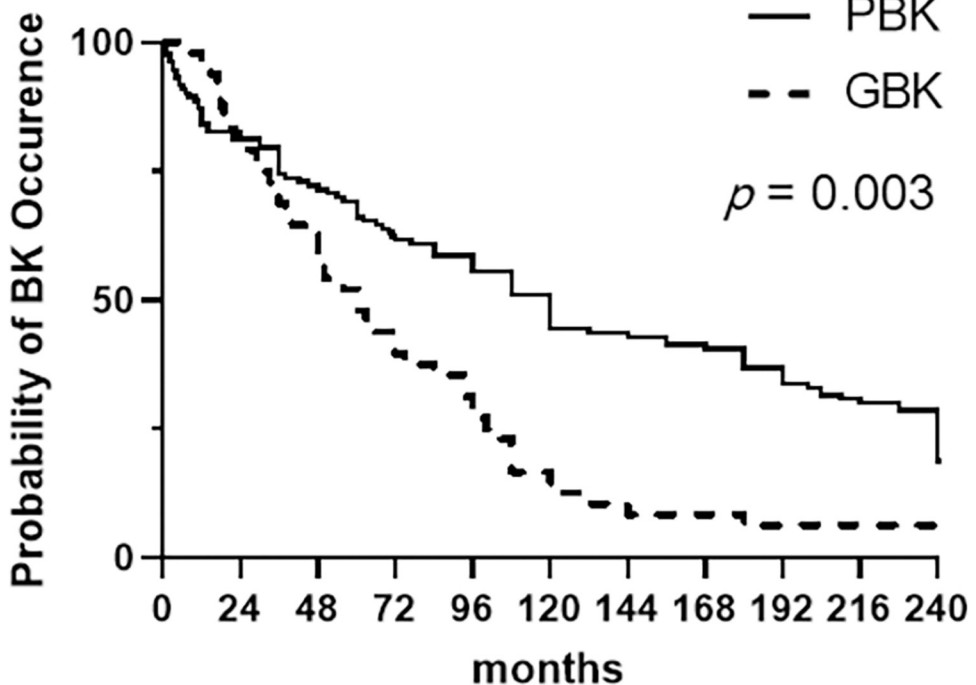

**Fig 2. Kaplan-Meier survival curves for the median time to BK onset following surgery in eyes with PBK and GBK.** The median BK onset times were 60.0 months in GBK and 120.0 months in PBK ($p = 0.003$ as analyzed by the log-rank test).

## Outcome of PK

Totally, PK was performed in 114 eyes (33.5%) among 340 BK eyes. Among 114 eyes, 5 eyes patients with < 6 months of the post-PK follow-up were excluded (two PBK eyes, two GBK eyes and one uveitis-induced BK), and the remaining 109 eyes were included for further outcome analysis. The overall graft success rate was 43.1% (47 out of 109 eyes) for 53.9 ± 29.9 months of post-PK follow-up as determined by graft clarity at last follow-up. The mean and median survival times to graft failure were 49.1 ± 4.1 months and 42 months, respectively. The graft survival rates and times according to BK etiologies are presented in Table 1.

## Comparison of PK outcome between PBK and GBK

Our results indicated that cataract surgery and glaucoma surgery/laser were the leading causes of BK in our population, representing 47.7% and 20.6% of total 340 BK eyes, respectively (Fig 1, Table 1). Similarly, PBK and GBK were identified to be the top two indications for PK in BK eyes, accounting for 44.7% and 12.3% of total 114 PK cases, respectively (Table 1). Thus, we next analyzed and compared the outcomes of PK between PBK and GBK.

Table 2 displays demographical data, pre-PK clinical characteristics and post-PK outcomes in eyes with PBK (49 eyes of 49 patients) and GBK (12 eyes of 12 patients) who received PK for the treatment of BK and were followed-up for > 6 months after PK. Among the 49 PBK eyes, 35 had undergone PE, 7 had undergone ECCE, and 6 had undergone ICCE, while the type of cataract surgery was unknown for one eye. Also, 43 out of 49 PBK eyes had PC IOL implantation, while 6 eyes had AC IOL. Among the 12 GBK eyes, 11 had undergone GDD surgery, and one had undergone trabeculectomy. There were no significant differences in age, gender, laterality of an involved eye, general medical history, pre-PK VA, pre-PK IOP and follow-up duration between PBK and GBK patients (Table 2). All eyes in both GBK and PBK groups had normal, well-controlled IOP before PK. The time from glaucoma surgery to GBK onset (60.8 ± 31.9 months) was significantly shorter than the time from cataract surgery to PBK (155.9 ± 128.3 in PBK, $p < 0.001$). The interval between surgery to PK was markedly shorter in GBK than in PBK (74.2 ± 33.7 months vs 162.0 ± 131.4 months, $p < 0.001$).

After uneventful PK in all eyes, the mean survival time was shorter in GBK eyes (28.3 ± 5.9 months) than in PBK eyes (56.4 ± 6.6 months, $p = 0.020$) (Table 2). Similarly, the median survival time of corneal allografts, as analyzed by Kaplan-Meier survival curves, was significantly shorter in GBK group (24.0 months), compared with PBK group (51.0 months, $p = 0.020$) (Fig 3 and Table 2). Specifically, the graft survival rates in GBK and PBK groups were 83.3% and 93.9% at 1 year ($p = 0.252$), 36.4% and 80.0% at 2 years ($p = 0.008$), and 30.0% and 57.6% at 3 years after PK ($p = 0.162$), respectively (Fig 3).

The most common cause of graft failure in GBK group was an immunologic endothelial rejection (Table 2). The rejection occurred more frequently in GBK eyes (6 of 12 eyes, 50%) than in PBK eyes (7 of 49 eyes, 14.3%) ($p = 0.014$). Chronic endothelial decompensation (in the absence of clinically evident rejection) was observed in in 2 of 12 (16.7%) GBK eyes and in 22 of 49 (44.9%) PBK eyes ($p = 0.003$). Graft infection developed in 2 eyes (4.1%) of PBK group at 14 and 24 months after PK, respectively, while no infection was observed after PK in GBK group. Secondary glaucoma (IIOP after PK) occurred in 14 of 49 (28.6%) PBK eyes, of which 6 eyes required GDD implantation. IOP was well-controlled after PK in 11 of 12 (91.7%) GBK eyes except in one eye experiencing IIOP after PK. Otherwise, there were no complications such as wound leak, hypotony, choroidal detachment, endophthalmitis or retinal detachment in either GBK or PBK eyes.

**Table 2. Comparison of PK outcomes between GBK and PBK.**

| | GBK (n = 12) | PBK (n = 49) | *P* value |
|---|---|---|---|
| Pre-PK characteristics | | | |
| Age | 60.1 ± 12.9 | 61.9 ± 13.4 | 0.681 |
| Gender (M: F) | 9: 3 | 33: 16 | 0.737 |
| Laterality (R: L) | 6: 6 | 18: 31 | 0.513 |
| Diabetes mellitus (n, %) | 1 (8.3) | 8 (16.3) | 0.673 |
| Mean OT (month) | 60.8 ± 31.9 | 155.9 ± 128.3 | <0.001 |
| Mean interval from event to PK (months) | 74.2 ± 33.7 | 162.0 ± 131.4 | <0.001 |
| Mean interval from BK to PK (months) | 13.3 ± 10.3 | 6.0 ± 28.9 | 0.395 |
| Pre-PK BCVA (logMAR) | 1.9 ± 0.5 | 1.8 ± 0.4 | 0.582 |
| Pre-PK IOP (mmHg) | 15.4 ± 3.7 | 14.1 ± 4.4 | 0.351 |
| Lens status (phakia: pseudophakia) | 3: 9 | 0: 49 | 0.006 |
| Glaucoma diagnosis | 12 (100) | 13 (26.5) | <0.001 |
| No. of glaucoma drops | 1.9 ± 1.3 | 0.5 ± 1.0 | <0.001 |
| Mean ST (months) | 28.3 ± 5.9 | 56.4 ± 6.6 | 0.020 |
| Median ST (months) | 24.0 | 51.0 | 0.020 |
| Post-PK BCVA (logMAR) | | | |
| Improved BCVA (n, %) | 9 (75.0) | 38 (77.6) | 0.385 |
| Postoperative 6 months | 1.2 ± 0.6 | 1.0 ± 0.5 | 0.377 |
| Postoperative 1 year | 1.4 ± 0.7 | 0.9 ± 0.6 | 0.017 |
| Postoperative 2 year | 1.6 ± 0.7 | 1.1 ± 0.8 | 0.110 |
| Postoperative 3 year | 1.8 ± 0.7 | 1.1 ± 0.8 | 0.043 |
| Post-PK ECD | | | |
| Postoperative 6 months | 1544 ± 751 | 1933 ± 818 | 0.240 |
| Postoperative 1 year | 846 ± 653 | 1477 ± 886 | 0.113 |
| Postoperative 2 year | 588 ± 248 | 1190 ± 889 | 0.265 |
| Post-PK CCT | | | |
| Postoperative 6 months | 556 ± 50 | 540 ± 50 | 0.462 |
| Postoperative 1 year | 583 ± 74 | 539 ± 80 | 0.268 |
| Postoperative 2 year | 750 ± 233 | 545 ± 84 | 0.264 |
| Post-PK complication | | | |
| Infection (n, %) | 0 (0) | 2 (4.1) | 1.000 |
| Elevated IOP (n, %) | 1 (8.3) | 14 (28.6) | 0.262 |
| Endothelial rejection (n, %) | 6 (50) | 7 (14.3) | 0.014 |
| Chronic endothelial decompensation (n, %) | 2 (16.7) | 22 (44.9) | 0.003 |
| Post-PK FU duration (months) | 50.0 ± 33.9 | 57.9 ± 29.9 | 0.426 |

GBK: glaucoma surgery-associated bullous keratopathy, PBK: pseudophakic bullous keratopathy, PK: penetrating keratoplasty, OT: onset time of BK from cataract or glaucoma surgery, BCVA: best-corrected visual acuity, ST: graft survival time, ECD: endothelial cell density, CCT: central corneal thickness, IOP: intraocular pressure, FU: follow-up

The improvement in BCVA was achieved after PK in most of GBK and PBK eyes (Table 2). In line with annual graft survival rates, however, BCVA (logMAR) was significantly lower in GBK eyes than in PBK eyes at 1 and 3 years after PK (1.4 ± 0.7 vs 0.9 ± 0.6, $p$ = 0.017 at 1 year; 1.8 ± 0.7 vs 1.1 ± 0.8, $p$ = 0.043 at 3 years). ECD, as measured in corneal grafts by specular microscopy, were consistently lower during follow-up, while CCT thickness measured by pachymetry was consistently thicker, in GBK eyes compared to PBK eyes, but the differences did not reach statistical significance (Table 2).

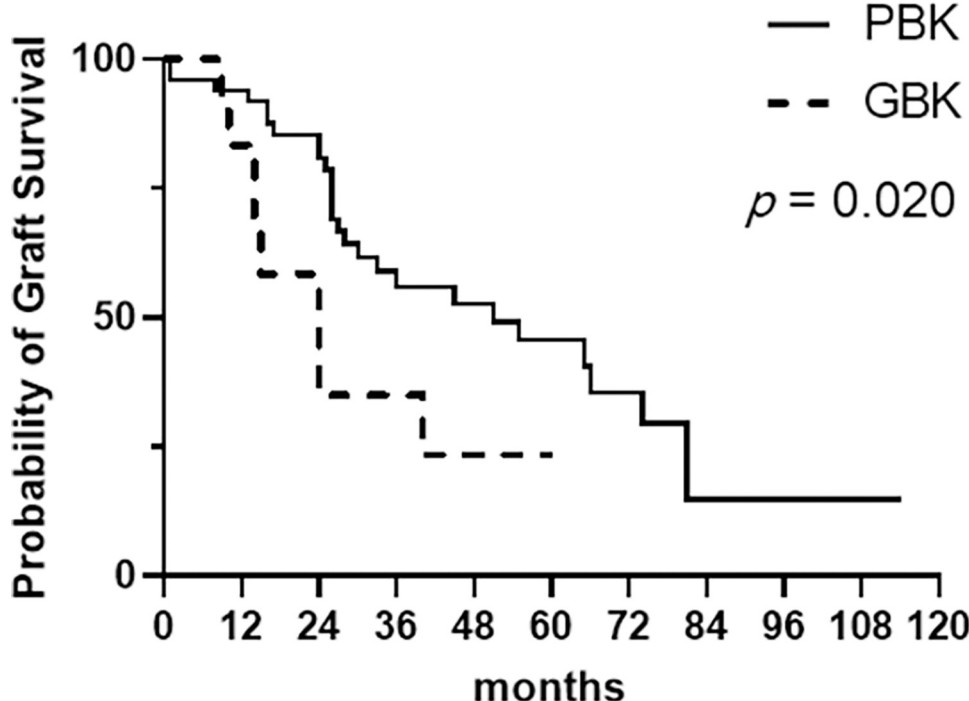

**Fig 3. Kaplan-Meier survival curves for corneal allotransplants in eyes with PBK and GBK.** The median survival times of corneal transplants were 24.0 months in GBK and 51.0 months in PBK ($p$ = 0.020 as analyzed by the log-rank test).

### Factors affecting PK outcome in PBK and GBK

We went on to seek for risk factors affecting PK outcome in GBK and PBK eyes. Multivariate Cox proportional hazards regression models confirmed that pseudophakic eyes had lower relative risk of graft failure (0.20, 0.05 ─ 0.75, $p$ = 0.017) (Table 3). These results of multivariate analysis are consistent with the results of direct comparisons of graft survival times and rates between GBK and PBK (Table 2). Among the postoperative factors, immunologic endothelial rejection (2.72, 1.22 ─ 6.06, $p$ = 0.014) and graft infection (10.29, 2.14 ─ 49.33, $p$ = 0.004) were

**Table 3. Cox's proportional hazard regression analysis for risk factors associated with PK outcome.**

| Variables | GBK (n = 12) | | | PBK (n = 49) | | | Total (n = 61) | | |
|---|---|---|---|---|---|---|---|---|---|
| | HR | 95% CI | *P* value | HR | 95% CI | *P* value | HR | 95% CI | *P* value |
| Age | 0.86 | 0.72–1.03 | 0.105 | 0.98 | 0.96–1.01 | 0.296 | 0.98 | 0.95–1.00 | 0.069 |
| Gender | 0.001 | 0.00–8.14e142 | 0.969 | 1.17 | 0.45–3.04 | 0.749 | 1.13 | 0.54–3.11 | 0.562 |
| Pre-PK | | | | | | | | | |
| Pseudophakia | 0.75 | 0.01–2.26 | 0.747 | | | | 0.20 | 0.05–0.75 | 0.017 |
| GDD tube in AC | 1.06 | 0.13–8.91 | 0.961 | | | | 1.94 | 0.69–5.44 | 0.211 |
| Post-PK | | | | | | | | | |
| Immunologic rejection | 19.22 | 1.30–283.33 | 0.031 | 1.57 | 0.51–4.81 | 0.431 | 2.72 | 1.22–6.06 | 0.014 |
| IIOP | 651.70 | 0.00–4.07e148 | 0.970 | 0.98 | 0.39–2.51 | 0.974 | 0.91 | 0.38–2.19 | 0.838 |
| Infection | - | | - | 10.46 | 2.09–52.48 | 0.004 | 10.29 | 2.14–49.33 | 0.004 |

AC: anterior chamber, BK: bullous keratopathy, GBK: glaucoma surgery-associated bullous keratopathy, PBK: pseudophakic bullous keratopathy, HR: hazard ratio, CI: confidence interval, PK: penetrating keratoplasty, GDD: glaucoma drainage device, IIOP: increased intraocular pressure

significantly associated with higher relative risk of graft failure (Table 3). Other factors including age, gender, the presence of GDD tube in the AC or post-PK IIOP did not significantly affect the graft survival.

## Discussion

This study documented the causes of BK in the Korean population over the past 10 years and analyzed the outcomes of PK in patients with BK according to its etiology. In our series, cataract surgery and glaucoma surgery/laser were the most common conditions leading to BK, accounting for 47.7% and 20.6% of total BK cases, respectively. These results reflect the importance of intraocular surgeries in the development of BK in our population, and are in accordance with reports from other Asian countries such as China and Japan [6, 11].

The incidence of BK after cataract surgery is estimated to be 0.6% to 2% in patients undergoing the surgery [12, 13], while the incidence of BK following glaucoma surgery/laser has not been reported. As the population ages, the incidence of cataract and glaucoma, which require surgical intervention, is increasing. Moreover, the number of glaucoma surgery is rising every year, and GDD implantation has become an important method for lowering of IOP in glaucoma patients [14, 15]. It can be presumed that BK associated with cataract and glaucoma surgeries will become more prevalent in the future. Therefore, it is important to understand the clinical characteristics and therapeutic outcomes of BK following cataract and glaucoma sugeries.

In this study, we found that BK following cataract surgery is characterized by late onset as compared to BK associated with other etiologies. For example, PBK developed at the mean 148.4 months (the median 120 months) after surgery, whereas GBK developed earlier at the mean 85.0 months (the median 60 months) following glaucoma surgery. BK developed even faster at the mean 7.8 months following vitrectomy. During cataract surgery, the most common cause of post-operative corneal edema is surgical injury to the corneal endothelium induced by ultrasound energy, turbulence of the irrigating solution, ricocheting of nuclear fragments and contact with surgical instruments [5, 13]. Therefore, it should be considered that surgical trauma and IOL might be causes of corneal decompensation occurring many years after cataract surgery, especially in eyes with shallow chamber and narrow angle. Moreover, in GBK eyes, corneal endothelial cells were damaged by the long-standing glaucoma and additionally injured by glaucoma surgery, which might be a reason for faster onset of BK in GBK eyes compared to PBK only eyes. Given these, surgical technique modification, alongside intraoperative and postoperative care, would help to reduce the risk of corneal endothelial decompensation in patients following surgery. In the same vein, a recent introduction of minimally invasive glaucoma surgery, called MIGS [16], may contribute to further reduction of the risk of BK in patients following glaucoma surgery.

Both univariate and multivariate analyses in our study revealed that GBK, compared to PBK, was associated with lower graft survival and poorer visual outcome after PK. These results are consistent with previous reports showing that the risk of corneal graft failure is significantly increased in glaucoma eyes with prior glaucoma surgery or using preoperative glaucoma medications [17–19]. Another intriguing finding of our study is that graft failure was more prevalent in GBK despite well-controlled IOP; IOP was increased in 8% of GBK eyes after PK, whereas 29% of PBK eyes had IIOP after PK. Importantly, in GBK eyes, immunologic endothelial rejection was the most common cause of graft failure; 50.0% of GBK eyes had an immune rejection after PK, while 14.3% of PBK eyes had the rejection. This finding suggests that GDD-implanted eyes might be vulnerable to an immunologic endothelial rejection presumably through exposure of intraocular antigens to systemic immune system [20, 21]. Hence,

our data emphasize the notion that immune reaction should be monitored cautiously and controlled effectively in GBK eyes during the post-PK follow-up for prevention of graft failure.

There are several limitations in this study. First, there is a difference in the number of PBK and GBK eyes due to such a large difference in initial surgery volumes between cataract and glaucoma surgeries, rendering a direct comparison between the two groups difficult. Regardless, we believe that our data (incidence of BK accorging to etiology, onset time of BK from insulting incident, and corneal graft failure rates and their survival times according to BK etiologies) would provide useful information on the importance of intraocular surgery-associated BK and its outcome in Asian eyes. Second, due to the retrospective nature of the study, detailed information on clinical and biological factors such as ECD and IOL position was often missing in medical charts, posing a challenge to further analysis of various individual risk factors. Third, the GBK group included the eyes that underwent cataract surgery as we defined GBK as BK following glaucoma surgery regardless of a history of cataract surgery, while we designated PBK as cases developing BK after cataract extraction and IOL implantation without history of glaucoma surgery or laser. This was inevitable because the majority of GBK patients had undergone cataract surgery in the real world, as was the case in our own case series and in other studies [22–24]. Thus, our results should be interpreted with caution, considering the possibility that glaucoma surgery may act as an aggravating factor in BK development rather than its isolated cause. Nevertheless, our data supports the notion that glaucoma surgery plays a critical role in BK development and PK failure.

## Conclusions

In conclusion, we herein present the recent trend in BK etiologies and PK indications in the Korean population, highlighting the importance of BK associated with intraocular surgery. Developing surgical techniques that minimally affect the corneal endothelium during and after surgeries would be beneficial in preventing BK development. Our data also revealed an earlier onset of BK following glaucoma surgery and poorer PK outcomes in GBK patients. This warrants future investigation into the pathogenesis of BK and corneal graft failure in eyes with glaucoma and glaucoma surgery. Additionally, improving corneal transplant surgical techniques and postoperative management can help enhance graft outcomes in BK eyes. With the new era of minimally invasive intraocular surgeries and endothelial keratoplasties, we can expect favorable changes in the incidence of BK and its therapeutic outcomes.

## Author Contributions

**Conceptualization:** Joo Youn Oh.

**Data curation:** Young-ho Jung, Joo Youn Oh.

**Formal analysis:** Young-ho Jung, Joo Youn Oh.

**Funding acquisition:** Joo Youn Oh.

**Investigation:** Young-ho Jung, Joo Youn Oh.

**Methodology:** Young-ho Jung, Joo Youn Oh.

**Project administration:** Joo Youn Oh.

**Resources:** Hyuk Jin Choi, Mee Kum Kim, Joo Youn Oh.

**Supervision:** Joo Youn Oh.

**Validation:** Joo Youn Oh.

**Writing – original draft:** Young-ho Jung.

**Writing – review & editing:** Hyuk Jin Choi, Mee Kum Kim, Joo Youn Oh.

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
