## [Decision Letter · Decision Letter 0]

20 Mar 2023

PONE-D-22-29033Etiology and outcome of penetrating keratoplasty in bullous keratopathy post-cataract surgery vs post-glaucoma surgeryPLOS ONE

Dear Dr. Oh,

Thank you for submitting your manuscript to PLOS ONE. After careful consideration, we feel that it has merit but does not fully meet PLOS ONE’s publication criteria as it currently stands. Therefore, we invite you to submit a revised version of the manuscript that addresses the points raised during the review process.

ACADEMIC EDITOR: The authors need to be commended for planning this study, however there is a lot of scope of improvement. Please see the comments suggested below.

We look forward to receiving your revised manuscript.

Kind regards,

Natasha Gautam, MBBS, MS

Academic Editor

PLOS ONE

“This work was supported by the National Research Foundation of Korea (2021R1A2C3004532 to J.Y.O).”

Please state what role the funders took in the study.  If the funders had no role, please state: ""The funders had no role in study design, data collection and analysis, decision to publish, or preparation of the manuscript.

Additional Editor Comments:

1. The authors should classify the type of cataract surgery. Was it clear corneal Phacoemulsification, extracapsular or intracapsular cataract extraction because it will have a bearing on Endothelial cell loss. The authors should mention the number of patients in each subcategory, if they included all 3 categories.

2. It is suggested that authors add another row in Table 1, representing total post cataract surgery (ABK + PBK) demographic variables, and also keep the current rows of AK and PBK demographic rows separately, so the readers can look at the tables to compare the PBK and AK group characteristics if they desire. Table 2 should also include 3rd column of Aphakic Bullous Keratopathy, besides discussing PBK and GBK, since ABK is important reason after cataract surgery, though it won't be fair to club PBK and ABK together. So the authors are requested to describe the 3 columns and compare them using appropriate statistical results.

3. Did the authors combine both GDD and Laser related cases in GBK in Table 2? If not, both of them should be combined, since they are post Glaucoma reasons for PBK.

4. It would be appreciated if authors could identify the underlying surgeries in GBK group, like how many underwent GDD, Trabeculectomy or trabecuolotomy.

5. In line 269, the authors stated that GDD is primary method of lowering IOP in glaucoma patients, which should be reframed, since its primary method in secondary glaucoma or post PK eyes, and not in all primary glaucoma patients.

6. The discussion would need revision for incorporating all the modifications suggested in results section.

Reviewers' comments:

Reviewer's Responses to Questions

**Comments to the Author**

1. Is the manuscript technically sound, and do the data support the conclusions?

Reviewer #1: Yes

Reviewer #2: Yes

2. Has the statistical analysis been performed appropriately and rigorously? 

Reviewer #1: No

Reviewer #2: I Don't Know

3. Have the authors made all data underlying the findings in their manuscript fully available?

Reviewer #1: Yes

Reviewer #2: Yes

4. Is the manuscript presented in an intelligible fashion and written in standard English?

Reviewer #1: Yes

Reviewer #2: Yes

5. Review Comments to the Author

Reviewer #1: It is a meaningful study. However, a more structured description is needed

1） In title, author focused on“post-cataract surgery vs post-glaucoma surgery”. But in manuscript, PBK is the main topic. (Line 23) . But aphakic BK also is one main BK cause by cataract surgery (Line 130), Both AK and PBK results should be described while describing post cataract surgery BK.    

2） Same for the glaucoma surgery/laser BK, (line 133). The LI and GBK should been combined in the post-glaucoma surgery catalog （ Line150， Table 1）.

3） For the cox’s hazard regression analysis, the tube and other implant devices in AC of GDD surgery should been discussed such as pseudophakia lenses in cataract surgery

4） The author hypothesized the causes for chronic corneal endothelium failure are surgical injury，but the tilt and dislocation IOL are the major reason for chronic PBK. The further investigation of IOL should been conducted.

5） In line 319, the author defined GBK as BK following glaucoma surgery irrespective of a history of cataract surgery. But how to evaluate the cataract surgery combined with Trab surgery, or combined with Tube surgery? These type surgeries should be presented as separate type.

Reviewer #2: The paper is a retrospective study so some important data was lacking which could have been collected if done prospectively. The authors describes their available data in well organized scientific way but there are still unanswered questions:

The definition of Glaucoma Bullous Keratopathy included patients irrespective of history of cataract surgery. The authors are suggested to provide a timeframe and more details about this inclusion criteria. Were the patients who underwent combined phacoemulsification with GDD/ Trabeculectomy included in the GBK group? If yes, these patients should be excluded from the study to avoid potential confounding.What all Laser procedures were included for the purpose of this study? Was it limited to laser iridotomy, or included other procedures like Laser cyclophotocoagulation, Laser trabeculoplasty?6 months follow up is a short term follow up. I would suggest to include this in titleWhat method was used to measure the IOP in post PK patients and what criteria were used to adjust the corneal factors?Were any specular microscopy studies done to document the rate of endothelial loss?**********

6. PLOS authors have the option to publish the peer review history of their article (what does this mean?). If published, this will include your full peer review and any attached files.

Reviewer #1: No

Reviewer #2: **Yes: **Ajay Jurangal

---

## [Author Response · Author response to Decision Letter 0]

4 Apr 2023

To the editor: 

=> Reply: We can confirm that our manuscript adheres to PLOS ONE's style requirements. 

2. Please provide additional details regarding participant consent. In the ethics statement in the Methods and online submission information, please ensure that you have specified (1) whether consent was informed and (2) what type you obtained (for instance, written or verbal, and if verbal, how it was documented and witnessed). If the need for consent was waived by the ethics committee, please include this information.

=> Reply: In the revised manuscript, we added a sentence regarding participant consent in the Methods (Line 64-66): “Due to the retrospective nature of the study, the IRB waived the requirement for obtaining informed consent from the patients.” 

3. Please state what role the funders took in the study.

=> Reply: As requested, we added the following sentence in the cover letter: “The funders had no role in study design, data collection and analysis, decision to publish, or preparation of the manuscript”.

4. The authors should classify the type of cataract surgery. Was it clear corneal Phacoemulsification, extracapsular or intracapsular cataract extraction because it will have a bearing on Endothelial cell loss. The authors should mention the number of patients in each subcategory, if they included all 3 categories.

=> Reply: In response to the editor’s request, we included the respective number of patients who underwent PE, ECCE or ICCE in the revised manuscript (Line 131-133, 137-139, 202-204). 

5. It is suggested that authors add another row in Table 1, representing total post cataract surgery (ABK + PBK) demographic variables, and also keep the current rows of AK and PBK demographic rows separately, so the readers can look at the tables to compare the PBK and AK group characteristics if they desire. Table 2 should also include 3rd column of Aphakic Bullous Keratopathy, besides discussing PBK and GBK, since ABK is important reason after cataract surgery, though it won't be fair to club PBK and ABK together. So the authors are requested to describe the 3 columns and compare them using appropriate statistical results.

=> Reply: As requested, we have added two rows in Table 1, one representing the data for ABK + PBK group and the other representing the data for GBK + LI, while retaining the rows of ABK, PBK, GBK and LI separately as they are (the revised Table 1). We agree with the editor and reviewers that this information would help a better understanding of our study’s findings. 

Regarding Table 2, we sought to compare the outcomes of penetrating keratoplasty (PK) between the two most common indications for PK in BK eyes: PBK and GBK, which accounted for 44.7% (n = 51) and 12.3% (n = 14) of the total 114 PK cases, respectively. We elaborated on this in the manuscript (Line 193-198). Unfortunately, due to the small sample size (only eight eyes) in the ABK group that had undergone PK, we were unable to reliably compare the outcomes of this group with those of the PBK or GBK groups. Instead, we presented descriptive data in Table 1, including the mean graft survival time, rate of graft failure, and follow-up period after PK, categorized by the causative condition of BK.

6. Did the authors combine both GDD and Laser related cases in GBK in Table 2? If not, both of them should be combined, since they are post Glaucoma reasons for PBK.

=> Reply: As stated in the Methods (Line 80-84), GBK was defined as BK following glaucoma surgery (GDD implantation, trabeculectomy and trabeculotomy), but not laser. Therefore, the data in the GBK group presented in Table 2 represents PK outcomes in BK patients who had undergone glaucoma surgery (not laser). As presented in Table 1, only 4 laser-related eyes had undergone PK. Due to the limited number of laser-related PK cases, we did not provide detailed analysis of PK outcomes in this group in Table 2. Instead, we presented descriptive data on PK outcome in Table 1, including the mean graft survival time, rate of graft failure, and follow-up period after PK, in the laser-related BK group. 

Agreeing with the editor's suggestion that it would be helpful to combine the data for GBK and laser-related BK, we added a new row in the revised Table 1, representing the demographics, clinical characteristics and PK outcome for the combined group of GBK and laser-related BK (GBK + LI). 

7. It would be appreciated if authors could identify the underlying surgeries in GBK group, like how many underwent GDD, Trabeculectomy or trabeculotomy. 

=> Reply: As per the editor’s suggestion, the relevant information has been provided in Line 140-143 as follows: “The second leading cause of BK was glaucoma surgery/laser, representing 20.6% (n = 70) of total BK cases (n = 340); 14.1% (n = 48) occurred following glaucoma surgery (GDD implantation in 43 eyes, trabeculectomy in 3 and trabeculotomy in 2), and the other 6.5% (n = 22) followed laser iridotomy.”

8. In line 269, the authors stated that GDD is primary method of lowering IOP in glaucoma patients, which should be reframed, since its primary method in secondary glaucoma or post PK eyes, and not in all primary glaucoma patients.

=> Reply: As advised, we rephrased the sentence in the Discussion as follows (Line 279-280): “GDD implantation has become an important method for lowering of IOP in glaucoma patients.”

9. The discussion would need revision for incorporating all the modifications suggested in results section. 

=> Reply: Appropriate modifications have been made throughout the manuscript.

To Reviewer #1: 

1. In title, author focused on “post-cataract surgery vs post-glaucoma surgery”. But in manuscript, PBK is the main topic. (Line 23). But aphakic BK also is one main BK cause by cataract surgery (Line 130), Both AK and PBK results should be described while describing post cataract surgery BK.

=> Reply: Fully agreeing with the reviewer, we described PBK and ABK separately when depicting the demographics of patients, the cause and onset of BK, and PK outcomes in Table 1 (Line 134-135, 173-174, Table 1). As for the comparative analysis of PK outcomes in Table 2 and Figure 3, we compared between PBK and GBK, the two most common indications for PK in BK eyes (Line 193-198). Due to the small sample size (only eight eyes) in the ABK group that underwent PK, we were unable to compare the PK outcomes of the ABK group with those of the PBK or GBK groups. Instead, we presented descriptive data in Table 1, including the mean graft survival time, rate of graft failure, and follow-up period after PK in the ABK group. Also, we added a new row in Table 1, representing the data for ABK + PBK group, while keeping the rows of ABK and PBK separately as they are (the revised Table 1).

2. Same for the glaucoma surgery/laser BK, (line 133). The LI and GBK should been combined in the post-glaucoma surgery catalog (Line150，Table 1). 

=> Reply: As per the reviewer’s suggestion, we have added a new row in Table 1 that represented the combined data for GBK + laser iridotomy (LI) cases, while retaining the separate rows for GBK and LI as they were. Please refer to the revised Table 1 for the updated data.

3. For the cox’s hazard regression analysis, the tube and other implant devices in AC of GDD surgery should been discussed such as pseudophakia lenses in cataract surgery. 

=> Reply: As per the reviewer’s suggestion, we conducted additional analysis to evaluate whether the presence of a GDD tube in the anterior chamber (AC) might be a potential risk factor affecting PK outcomes, using Cox’s proportional hazard regression analysis. Although the hazard ratio of AC tube for graft failure was 1.94, the p value was not statistically significant (p = 0.211). We have revised Table 3 to include these data and mentioned these findings in the revised manuscript (Table 3 and Line 263). 

4. The author hypothesized the causes for chronic corneal endothelium failure are surgical injury, but the tilt and dislocation IOL are the major reason for chronic PBK. The further investigation of IOL should been conducted.

=> Reply: Regrettably, we were unable to analyze the tilt and dislocation of IOL as a potential risk factor of PBK due to limited availability of such information in the medical records. Per the reviewer’s comment, we have added these aspects as one of limitations of the study in the Discussion (Line 326-328).

5. In line 319, the author defined GBK as BK following glaucoma surgery irrespective of a history of cataract surgery. But how to evaluate the cataract surgery combined with Trab surgery, or combined with Tube surgery? These type surgeries should be presented as separate type.

=> Reply: As the reviewer pointed, we defined GBK as BK following glaucoma surgery irrespective of a history of cataract surgery, while we defined PBK as BK following cataract surgery but without history of glaucoma surgery/laser. Thus, in cases where a patient had undergone both surgeries, they were designated and analyzed as GBK (Line 83-84). This was inevitable because, considering the patients’ age and time elapsed from surgery to BK, the majority of GBK patients had undergone cataract surgery in the real world as was the case in our own case series and in other studies (Ref 22-24). To address the reviewer’s concern, we emphasized in the Discussion section the need to interpret our study's results with caution, considering the possibility that glaucoma surgery may act as an aggravating factor in BK development rather than the sole cause of BK (Line 328-338). Nonetheless, we humbly submit that our study's results demonstrate the significance of intraocular surgery in BK development in Asian eyes and poorer outcomes in eyes receiving glaucoma surgery, in terms of BK onset and PK failure. Therefore, our study's findings have clinical implications, as they warrant vigilance in seeking better postoperative care and minimally-invasive surgical techniques.

To Reviewer #2: 

1. The definition of Glaucoma Bullous Keratopathy included patients irrespective of history of cataract surgery. The authors are suggested to provide a timeframe and more details about this inclusion criteria. Were the patients who underwent combined phacoemulsification with GDD/ Trabeculectomy included in the GBK group? If yes, these patients should be excluded from the study to avoid potential confounding.

=> Reply: As stated in the Methods (Line 77-84), we defined GBK as BK following glaucoma surgery regardless of a history of cataract surgery, while we defined PBK as BK following cataract surgery but without history of glaucoma surgery/laser. As such, BK cases that received both cataract and glaucoma surgeries were designated and analyzed as GBK in our study (Please note that patients in GBK and PBK groups in our study do not overlap, and in that sense two groups are mutually exclusive, rendering the statistical methods used in the study appropriate). 

We share the reviewer’s concern: the fact that some patients in GBK group also underwent cataract surgery might have confounded the results. However, this was inevitable because, considering the patients’ age and time elapsed from surgery to BK, the majority of GBK patients had undergone cataract surgery in the real world, as was the case in our own case series and in other studies (Ref 22-24). To address the reviewer’s concern, we clarified our criteria for PBK and GBK in the Methods (Line 77-84), and emphasized in the Discussion the need to interpret our results carefully with the understanding that glaucoma surgery might act as an aggravating factor of BK rather than the sole cause of BK (Line 328-338). Nevertheless, we humbly submit that our results demonstrate the importance of intraocular surgery in BK development in Asian eyes, as well as poorer outcomes in eyes that have undergone glaucoma surgery in terms of BK onset and PK failure. These findings have clinical implications, as they underscore the need for better postoperative care and minimally-invasive surgical techniques. 

2. What all Laser procedures were included for the purpose of this study? Was it limited to laser iridotomy, or included other procedures like Laser cyclophotocoagulation, Laser trabeculoplasty?

=> Reply: As mentioned in Line 143, all laser-related BK cases (n = 22) were caused by laser iridotomy.

3. 6 months follow up is a short term follow up. I would suggest to include this in title

=> Reply: As mentioned in Line 85, our inclusion criteria for PK cases required a minimum of 6 months of post-PK follow-up, but this does not mean that we evaluated patients at 6 months post-PK. As shown in Table 1, the mean post-PK follow-up period for all PK cases was 53.9 months (SD 29.9 months). Therefore, we do not believe that it is appropriate to include “6 months” in the title.

4. What method was used to measure the IOP in post PK patients and what criteria were used to adjust the corneal factors?

=> Reply: In post-PK patients, the IOP was measured using Goldmann applanation tonometry, rebound tonometry or non-contact tonometry. To adjust for corneal factors, the central corneal thickness and corneal curvature were taken into account using correction formulas such as the Ehlers and Doughty formulas. The determination of increased IOP and patient care were performed by glaucoma specialists. We have added these details to the revised manuscript (Line 97-101, new Ref. 9-10). Thank you for brining up an important point and giving us the opportunity to clarify it.

5. Were any specular microscopy studies done to document the rate of endothelial loss?

=> Reply: Specular microscopy was performed on patients that received PK, and the data have been presented in Table 2 and Line 243-246. However, the endothelial cell density data in the overall BK patients were inconsistently available. In order to accurately track the rate of endothelial cell loss in the overall BK patients and compare it between different causes of BK, a prospective study will be needed in the future. We mentioned this in the Discussion section (Line 326-328).

---

## [Editor Report · Decision Letter 1]

24 Apr 2023

Etiology and outcome of penetrating keratoplasty in bullous keratopathy post-cataract surgery vs post-glaucoma surgery

PONE-D-22-29033R1

Dear Dr. Oh,

We’re pleased to inform you that your manuscript has been judged scientifically suitable for publication and will be formally accepted for publication once it meets all outstanding technical requirements.

Kind regards,

Natasha Gautam, MBBS, MS

Academic Editor

PLOS ONE

Additional Editor Comments (optional):

The authors have appropriately addressed all the comments
---

## [Editor Report · Acceptance letter]

28 Apr 2023

PONE-D-22-29033R1 

Etiology and outcome of penetrating keratoplasty in bullous keratopathy post-cataract surgery vs post-glaucoma surgery 

Dear Dr. Oh:

I'm pleased to inform you that your manuscript has been deemed suitable for publication in PLOS ONE. Congratulations! Your manuscript is now with our production department. 

Kind regards, 

on behalf of

Dr. Natasha Gautam 

Academic Editor

PLOS ONE